# Peaking Industrial Energy-Related CO$_2$ Emissions in Typical Transformation Region: Paths and Mechanism

**Zhiyuan Duan** [1,2], **Xian'en Wang** [1,2], **Xize Dong** [1,2], **Haiyan Duan** [1,2,*] **and Junnian Song** [1,2,*]

[1] Key Lab of Groundwater Resources and Environment, Ministry of Education, Jilin University, Changchun 130021, China; duanzy18@mails.jlu.edu.cn (Z.D.); wxen@jlu.edu.cn (X.W.); dongxz19@mails.jlu.edu.cn (X.D.)

[2] College of New Energy and Environment, Jilin University, Changchun 130021, China

[*] Correspondence: duanhy1980@jlu.edu.cn (H.D.); songjunnian@jlu.edu.cn (J.S.); Tel.: +86-431-8516-8429

**Abstract:** Reducing CO$_2$ emissions of industrial energy consumption plays a significant role in achieving the goal of CO$_2$ emissions peak and decreasing total CO$_2$ emissions in northeast China. This study proposed an extended STIRPAT model to predict CO$_2$ emissions peak of industrial energy consumption in Jilin Province under the four scenarios (baseline scenario (BAU), energy-saving scenario (ESS), energy-saving and low-carbon scenario (ELS), and low-carbon scenario (LCS)). We analyze the influences of various factors on the peak time and values of CO$_2$ emissions and explore the reduction path and mechanism to achieve CO$_2$ emissions peak in industrial energy consumption. The results show that the peak time of the four scenarios is respectively 2026, 2030, 2035 and 2043, and the peak values are separately 147.87 million tons, 16.94 million tons, 190.89 million tons and 22.973 million tons. Due to conforming to the general disciplines of industrial development, the result in ELS is selected as the optimal scenario. The impact degrees of various factors on the peak value are listed as industrial CO$_2$ emissions efficiency of energy consumption > industrialized rate > GDP > urbanization rate > industrial energy intensity > the share of renewable energy consumption. But not all factors affect the peak time. Only two factors including industrial clean-coal and low-carbon technology and industrialized rate do effect on the peak time. Clean coal technology, low carbon technology and industrial restructuring have become inevitable choices to peak ahead of time. However, developing clean coal and low-carbon technologies, adjusting the industrial structure, promoting the upgrading of the industrial structure and reducing the growth rate of industrialization can effectively reduce the peak value. Then, the pathway and mechanism to reducing industrial carbon emissions were proposed under different scenarios. The approach and the pathway and mechanism are expected to offer better decision support to targeted carbon emission peak in northeast of China.

**Keywords:** industrial energy consumption; CO$_2$ emissions; reduction path; peak; STIRPAT model

## 1. Introduction

The Paris Agreement, which came into effect in 2015, requires countries to report on the long-term strategy for low-carbon development in 2050 [1]. Based on this, China proposes to achieve CO$_2$ emissions peak around 2030. As far as China is concerned, the proportion of energy consumption in the industrial sector has always been above 70% [2]. Therefore, all of the provinces are actively engaged in industrial CO$_2$ reduction to accomplish the target of CO$_2$ emissions peak of China. However, due to the different levels of economic development and industrial structure, the strategies of industrial CO$_2$ emissions reduction should vary among different provinces and regions in China.

In the process of China's industrial development, northeast China, as the base of heavy industry, made a distinguished contribution to the country's modernization before the 1970s. Today, however, the northeastern region's economic growth gradually falls behind that of the eastern coastal regions, because resource-based industries has been declining since 1990. Thus, the northeast region faces a dilemma of promoting social and economic development with industrial transformation. So, the strategies and models of industrial $CO_2$ reduction in these regions are obliged to be distinct from developed regions such as China's eastern coastal provinces. Otherwise, inappropriate carbon reduction strategies and models will impact the economic development of northeast China. In order to realize the strategies of $CO_2$ emission reduction in northeast China, this study explores the following two questions. What should be done to reduce industrial $CO_2$ emissions in such areas? How do such areas achieve the $CO_2$ emissions peak of industrial energy consumption? To solve these two questions, first, this research should analyze the influencing factors of industrial $CO_2$ emissions and quantify their impact on the peak of $CO_2$ emissions. Second, this research should propose paths to reduce $CO_2$ emissions in such typical regions and study the $CO_2$ emission reduction mechanism of industrial energy consumption.

Existing literature shows that $CO_2$ emission researches in the industrial field mainly focus on the analysis of influencing factors and the peak prediction of $CO_2$ emissions. For the first question, using Kaya identity, LMDI, STIRPAT and SDA model, numerous previous studies elucidate that the influencing factors of industrial $CO_2$ emissions including economic scale, industry structure, energy structure, energy intensity, population, urbanization and openness. Liu et al. [3] indicated that the economic output, R&D intensity, investment intensity, and energy structure were the driving factors for the increase of carbon emissions in the whole period applying LMDI approach to decompose industrial energy-related carbon emissions into eight factors during 2001–2015 for Henan Province. Wu et al. [4] divided the industrial carbon emission effected into five categories for 39 industrial sectors in Inner Mongolia from 2003 to 2012. The result reveals that the energy intensity effect was the crucial drivers of the decrease of carbon emissions while the industrial growth effect and population effect, critical driving forces, increase industrial sector carbon emissions. Lin et al. [5] and Cui et al. [6] suggested that industrial-scale was the leading force of carbon emission increase, while energy intensity had a negative impact on carbon emission, through Kaya identity and LMDI method in China's major energy-intensive industries. Zhang et al. [7], in Anhui, applying STIRPAT model to decompose industrial carbon emissions into five factors such as out-of-province investment, energy structure, per capita income, industrial structure and actual utilization of foreign capital during 2000–2015, proved that energy consumption structure was the main factor to increase carbon emissions. Liu et al. [8] proposed an extended STIRPAT model to investigate the six factors driving industrial carbon emissions in China. The results suggested that the direct effects of energy intensity and energy structure were positive to increase industrial carbon emissions, and the direct effects of investment scale, industrial economy, energy price and openness are non-significant. Deng et al. [9], using the SDA model to find the drivers behind the industrial energy-related carbon emission changes during the years 1997–2012 in Yunnan, illustrated that the sharp increase in exports of high-carbon products from metal processing and electricity sectors rose carbon emissions. According to existing research, influencing factors such as population, economic output, investment intensity, and energy structure are the drivers for the increase of $CO_2$ emissions, and only the positive effect of energy intensity on $CO_2$ emissions reduction is almost unanimously recognized. However, previous studies have not addressed how the factors drive the peak volume and peak times of $CO_2$ emissions.

For the second question, the studies could be classified into two fields, namely, predicting $CO_2$ emission peak, and analyzing the scenarios of peak. In the field of predicting $CO_2$ emission peak, based on historical data, some scholars estimated the trend of industrial carbon emissions via different predicting methods to testify the existence of the emission peak within the planned target. For example, Yu et al. [10] analyzed China's emission trajectory with different methods and proposed a new economic-carbon emission-employment multi-objective optimization model. The results of the model indicated that China's industrial energy-related carbon emissions would peak between 2022

and 2025. In the field of analyzing the scenarios of peak, some researchers simulated the $CO_2$ emission peak by setting various scenarios for future development on the premise that carbon emission peak exists. Zhang et al. [11] introduced influencing factors into the decomposition model and combined dynamic Monte Carlo simulation with scenario analysis to identify whether and how the 2030 carbon emission-reduction targets would be realized from China's industrial sector perspective. The results revealed that three scenarios in the research would realize the 2030 emission-peak target. Zhu et al. [12] analyzed the driving factors of industrial energy-related $CO_2$ emissions of 17 cities in China's Yangtze River Delta region from 2005 to 2014 using LMDI and predicted the carbon emission-reduction potential of these cities from 2015 to 2020. The results showed that there would be a carbon emissions peak in the moderate and aggressive scenarios. Based on the analysis of the $CO_2$ emissions caused by industrial energy consumption in Tianjin from 2005 to 2012, Ge et al. [13] employed the STIRPAT model, logistic regression model, and grey model to predict the carbon emissions in the future respectively. According to existing research, diverse models were utilized to predict the industrial carbon emission peak, and various scenarios were analyzed to reveal the scenario situation of different peaks and the $CO_2$ emission reduction measures in this scenario. However, previous studies lack a further discussion on the pathway of transitions between scenarios.

Taking northeast China as the research area, some scholars have carried out relevant studies to analyze the influencing factors of $CO_2$ emissions in old industrial bases, $CO_2$ emissions peak prediction, and $CO_2$ emissions reduction countermeasures. Chen et al. [14], showed that the increase of carbon emissions was primarily launched by growth in per-capita final demand in Heilongjiang during 2002–2012, adopting the SDA. Wen et al. [15] found that economic growth, investment structure, investment dependence, and energy structure are the main factors to promote the growth of industrial energy Carbon emissions in Liaoning via the LMDI model. In addition, our research group took Jilin Province as an example to do some relevant researches. In 2016, based on the application of the LEAP system, we predicted that the peak time of carbon emissions would appear in 2045, 2040, 2035 and 2025 via setting four scenarios to predict energy-consumption carbon emissions [16]. In 2019, we took advantage of the LEAP system to predict carbon emissions peak and analyze the reduction pathway in buildings during operation in Jilin Province [17]. The above research is the foundation of this paper.

Overall, what should be done to achieve the $CO_2$ emissions peak of industrial energy consumption by reducing the $CO_2$ emissions in typical transformation regions such as northeast China? The question has not been well studied and solved in previous studies. In order to fill in the gap, this study empirically explored the main driving path and mechanism of $CO_2$ emissions peak in industrial energy consumption from the perspective of peak by associating taking Jilin Province as the target area from 1995 to 2015. Based on the results of the analysis, we provide practical and effective recommendations for reducing industrial $CO_2$ emissions. The remainder of the paper is organized as follows: Section 2 describes the method and data. Section 3 describes the scenarios and coefficient settings of variables. The results and discussion of the empirical case study are provided in Section 4. The conclusions are drawn in Section 5.

## 2. Study Method and Data

The technology roadmap of this study is shown in Figure 1.

### 2.1. STIRPAT Model Extension

The STIRPAT model, a kind of extended stochastic environmental impact assessment model, proposed and established by Dietz T [18] remedies the defect that the IPAT equation only affects the proportional change in the dependent variable to the greatest extent. Statistical analysis and random estimation of the influencing factors are carried out through the non-linear model to ensure the accuracy of the results. Besides, this model can also be reasonably and effectively interfaced with the scenario

parameter settings used for prediction, thus facilitating the quantitative discussion of $CO_2$ emissions reduction paths. The model can be expressed as follow:

$$I = a \cdot P^b \cdot A^c \cdot T^d \cdot e \tag{1}$$

where *I* represents the environmental impact, *P* is the population, *A* is the economic development, *T* is the technical level, $\alpha$ is the model coefficient, *b*, *c*, *d* are the index of independent variables, *e* is the model error term.

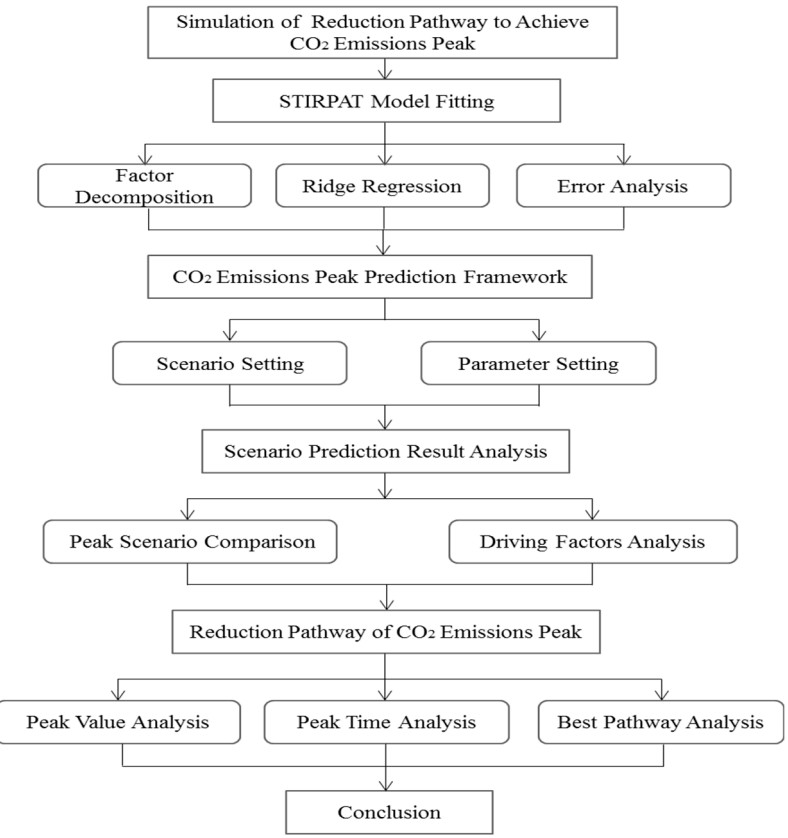

**Figure 1.** Study technology roadmap.

According to the relevant research results [19–21], based on the concept of residual-free decomposition of Kaya identity established by Yoichi Kaya, this study decomposes the influencing factors under the framework of IPAT model, which can be expressed as follow:

$$C_{int} = \frac{C_{int}}{E_{int}} \cdot \frac{E_{int}}{GDP_{int}} \cdot \frac{GDP_{int}}{GDP} \cdot GDP \tag{2}$$

where $C_{int}$ is $CO_2$ emissions of industrial energy consumption, $E_{int}$ is industrial energy consumption, $GDP_{int}$ is industrial GDP growth, *GDP* is the gross domestic product of the Jilin Province.

According to the Equation (1), the identity of industrial energy consumption $CO_2$ emissions can be expressed as follow:

$$C_{int} = a \cdot F_{int}^b \cdot T_{int}^c \cdot I_{int}^d \cdot A^e \cdot g \tag{3}$$

where $F_{int}$ is $CO_2$ emissions efficiency of industrial energy consumption, representing the ratio of $C_{int}$ and $E_{int}$, reflecting the level of clean-coal and low-carbon technology; $T_{int}$ is energy intensity of industry, representing the ratio of $T_{int}$ and $GDP_{int}$, which indicates the technological progress of industrial process and the level of energy efficiency; $I_{int}$ is rate of industrialization, representing the

ratio of $GDP_{int}$ and $GDP$, characterizing the level of industrialization; $A$ is $GDP$, characterizing the level of economic development.

Considering the effects of urbanization and clean energy utilization, this study adds the dimensionless factors of Urbanization rate (%) and the share of renewable energy (%) to further extend the STIRPAT model.

$$C_{int} = a \cdot F_{int}^b \cdot T_{int}^c \cdot I_{int}^d \cdot A^e \cdot Ps^f \cdot Es^g \cdot h \qquad (4)$$

where $P_S$ is the urbanization rate, characterizing the level of urbanization; $E_S$ is the share of renewable energy, characterizing the level of renewable energy use in total energy consumption. $e$, $f$ and $g$ are the indexes of independent variables. $h$ is the error term.

Taking the natural logarithm on both sides of the Equation (4) in order to eliminate the heteroscedasticity of the time series, the equation can be expressed as follow:

$$\ln C_{int} = \ln a + b(\ln F_{int}) + c(\ln T_{int}) + d(\ln I_{int}) + e(\ln A) + f(\ln Ps) + g(\ln Es) + \ln h \qquad (5)$$

The extended STIRPAT model includes six factors, such as industrial energy consumption $CO_2$ emission efficiency, industrial energy intensity, industrialization rate, GDP, urbanization rate and the share of renewable energy.

### 2.2. STIRPAT Model Ridge Regression Fitting

According to the extended STIRPAT model, Multivariate linear regression is performed with $\ln C_{int}$ as the dependent variable and "$\ln F_{int}$, $\ln T_{int}$, $\ln I_{int}$, $\ln A$, $\ln Ps$, $\ln Es$" as independent variables. Since the variance expansion factor (VIF) of multiple independent variables is much larger than 10, indicating that there may be multicollinearity among multiple variables, we choose the least-squares improved algorithm for equation fitting and add the nonnegative factor $k$ to improve the stability of the model estimation, that is, the ridge regression estimation method. In this study, applying the ridge regression macro program executes ridge regression fitting in SPSS, and the R squared and ridge trace corresponding to the estimated value of K were shown in Figures 2 and 3 respectively.

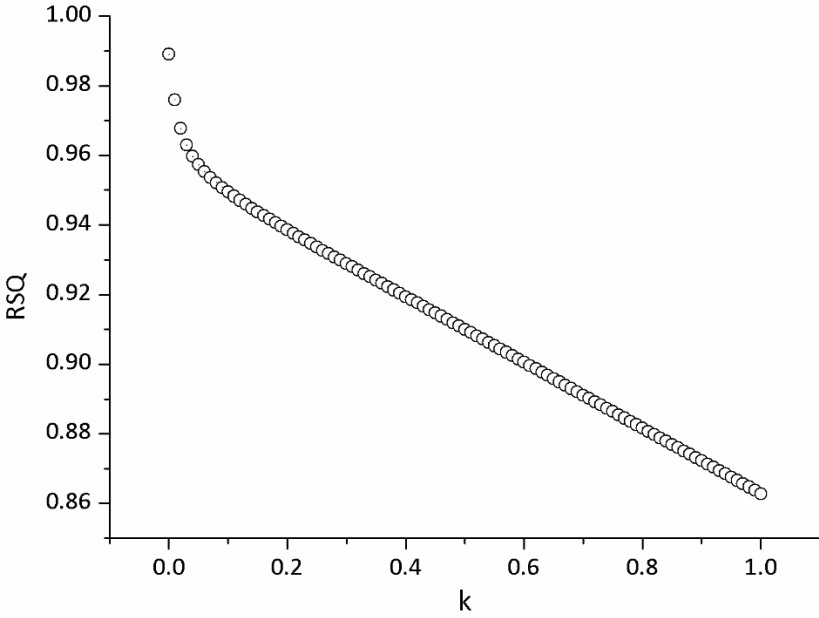

**Figure 2.** R-squared for estimated values of K.

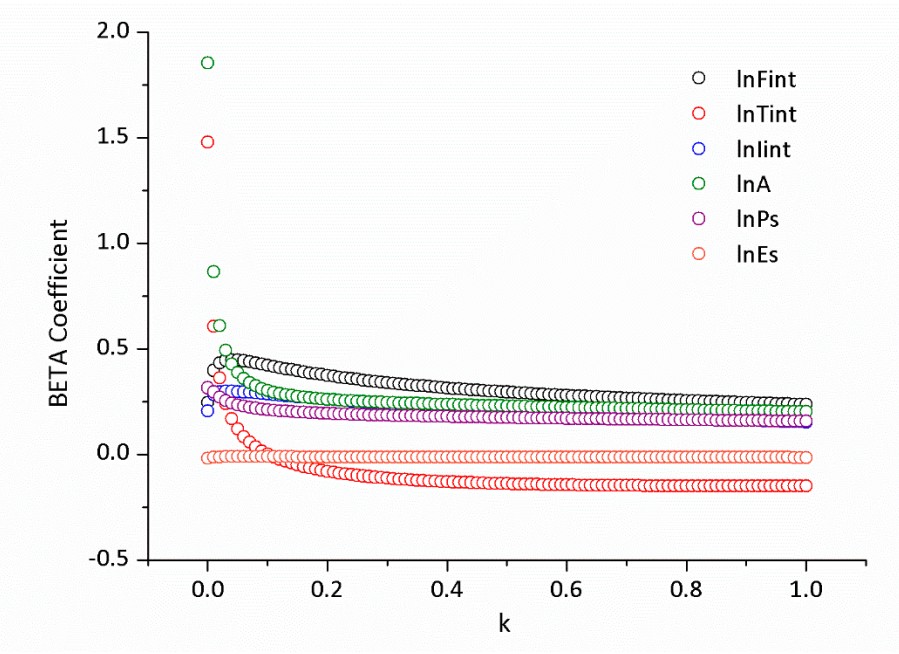

**Figure 3.** Ridge trace for estimated values of K.

When k = 0.45, the BETA coefficient of all influencing factors tends to be stable, and the adjusted R-squared is 0.91, indicating that the regression equation is significant, which is in accordance with the economic significance test. The corresponding ridge regression equation is expressed as follows:

$$\ln C_{int} = 1.38(\ln F_{int}) - 0.05(\ln T_{int}) + 0.83(\ln I_{int}) + 0.10(\ln A) + 0.39(\ln Ps) - 0.02(\ln Es) + 1.98 \quad (6)$$

In order to further verify the goodness of the ridge regression equation to ensure the stability and accuracy of the prediction results, the fitting data of $CO_2$ emissions ($\ln C_{int}'$) obtained in Equation (6) is compared with that of actual $CO_2$ emissions ($\ln C_{int}$). The margin of error is within 10%, as shown in Table 1, indicating that the fitting equation is available.

**Table 1.** Error statistics of actual $CO_2$ emissions and estimated values.

| Year | lnF$_{int}$ | lnT$_{int}$ | lnI$_{int}$ | LnA | lnPs | lnEs | lnC$_{int}'$ | lnC$_{int}$ | Error (%) |
|------|------|------|------|------|------|------|------|------|------|
| 1995 | 1.16 | 1.71 | 3.59 | 7.11 | 3.85 | 1.91 | 8.04 | 8.63 | 6.87 |
| 1996 | 1.17 | 1.57 | 3.56 | 7.24 | 3.85 | 1.90 | 8.10 | 8.62 | 5.98 |
| 1997 | 1.17 | 1.53 | 3.52 | 7.32 | 3.87 | 1.96 | 8.11 | 8.64 | 6.09 |
| 1998 | 1.12 | 1.29 | 3.46 | 7.41 | 3.88 | 1.87 | 8.11 | 8.42 | 3.63 |
| 1999 | 1.10 | 1.46 | 3.49 | 7.49 | 3.89 | 1.94 | 8.05 | 8.68 | 7.20 |
| 2004 | 1.08 | 1.01 | 3.51 | 7.58 | 3.91 | 1.91 | 8.61 | 8.39 | 2.64 |
| 2005 | 1.08 | 0.91 | 3.53 | 7.67 | 3.91 | 1.97 | 8.63 | 8.39 | 2.92 |
| 2006 | 1.08 | 0.88 | 3.53 | 7.76 | 3.93 | 2.06 | 8.73 | 8.47 | 3.07 |
| 2007 | 1.15 | 0.91 | 3.55 | 7.85 | 3.95 | 1.34 | 8.87 | 8.65 | 2.57 |
| 2008 | 1.17 | 0.79 | 3.60 | 7.97 | 3.96 | 1.36 | 9.00 | 8.73 | 3.06 |
| 2009 | 1.18 | 0.87 | 3.63 | 8.08 | 3.96 | 0.96 | 9.08 | 8.99 | 0.95 |
| 2010 | 1.20 | 0.80 | 3.66 | 8.22 | 3.97 | 1.10 | 9.30 | 9.12 | 1.94 |
| 2011 | 1.20 | 0.64 | 3.72 | 8.37 | 3.97 | 1.50 | 9.39 | 9.22 | 1.84 |
| 2012 | 1.23 | 0.53 | 3.73 | 8.52 | 3.97 | 1.69 | 9.43 | 9.32 | 1.21 |
| 2013 | 1.22 | 0.33 | 3.74 | 8.65 | 3.98 | 1.67 | 9.39 | 9.26 | 1.41 |
| 2014 | 1.27 | 0.15 | 3.81 | 8.78 | 3.98 | 1.80 | 9.43 | 9.33 | 1.10 |
| 2015 | 1.29 | 0.09 | 3.84 | 8.92 | 3.98 | 1.93 | 9.65 | 9.49 | 1.69 |

*2.3. Data*

The data were sourced from the Jilin Statistical Annual Book, covering GDP, industrial added value, industrialization rate, urbanization rate, energy consumption structure from 1995–2015. As for industrial energy consumption $CO_2$ emissions, this study utilizes the $CO_2$ emissions calculation method recommended by the Intergovernmental Panel on Climate Change (IPCC) to calculate $CO_2$ emissions using the physical quantities of energy consumption in the statistical yearbook. The descriptive statistics of the variables are presented in Table 2.

**Table 2.** The descriptive statistics of the variables.

| Variable | Obs. | Std. | Max | Mean | Min |
|---|---|---|---|---|---|
| $CO_2$ Emissions (Mt) | 21 | 33.61 | 132.59 | 84.35 | 44.13 |
| Industrial $CO_2$ emission efficiency (t$CO_2$/tce) | 21 | 0.20 | 3.63 | 3.25 | 2.92 |
| Industrial energy intensity (tce/ $10^4$ Yuan) | 21 | 1.42 | 5.52 | 2.40 | 0.73 |
| Industrialization rate (%) | 21 | 5.35 | 46.76 | 39.17 | 31.97 |
| GDP (Billion Yuan) | 21 | 313.63 | 1105.60 | 448.50 | 122.44 |
| Urbanization rate (%) | 21 | 2.52 | 55.11 | 51.62 | 46.98 |
| The share of renewable energy (%) | 21 | 1.65 | 8.41 | 6.09 | 2.60 |

## 3. Case Study

*3.1. Research Zone*

Northeast China mainly includes three provinces, Heilongjiang, Jilin, and Liaoning, with similar industry structure, energy structure, and technical level. According to the China Statistical Yearbook, the GDP of Heilongjiang, Jilin and Liaoning was, respectively, 1,537 billion, 1,478 billion and 2,225 billion Chinese yuan in 2017. The research area of this paper is Jilin Province, as shown in Figure 4, which has the lowest GDP among the three provinces. Based on the target of $CO_2$ emission reduction Jilin Province set, the $CO_2$ intensity would be decreased by 18.5% by 2020 [17]. $CO_2$ emissions from the industrial sector should be given priority because, in 2016, industrial energy consumption accounted for 64% of Jilin Province's total $CO_2$ emissions. STIRPAT model was utilized to determine the driving factors and forecast the peak of $CO_2$ emission from industrial energy consumption. On the ground of the analysis, the pathway meeting the reduction target in the old industrial base was presented.

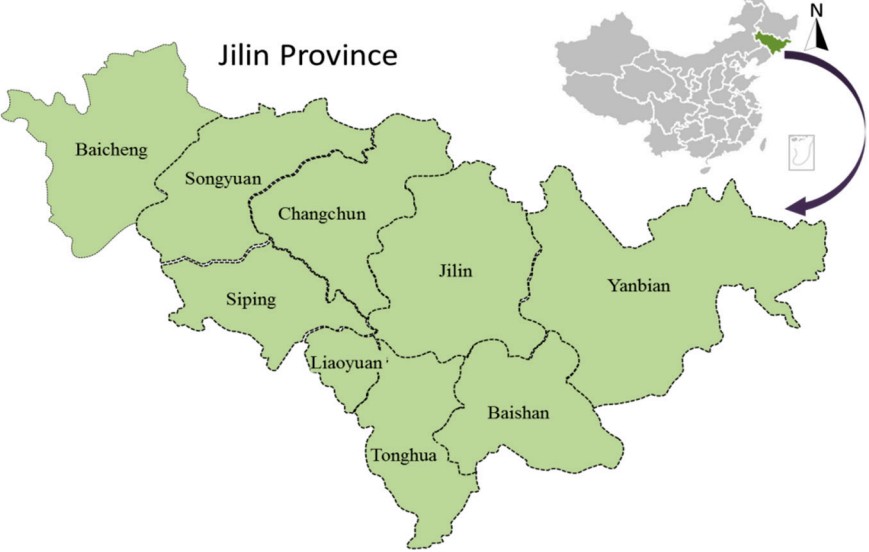

**Figure 4.** Map of study area.

*3.2. Scenario description*

Scenario analysis supplied various possible strategies with strict ratiocination and specific description based on the crucial assumptions on the development of society, economy, industry, and technology. According to the future development paths of Jilin Province, this study sets four scenarios, namely business as usual scenario, energy-saving scenario, energy-saving, and low-carbon scenario and low-carbon scenario. Because the countries mentioned in the Paris agreement need to report long-term strategies for low-carbon development before 2050, the scenarios of this study are set from 2016 to 2050.

3.2.1. Baseline Scenario (BAU).

In the baseline scenario (BAU), the development of various influencing factors continues the historical development status of Jilin Province before the 13th Five-Year Plan of Jilin Province. Based on the current situation, the impact of relevant policies and plans before the 13th Five-Year Plan on the future development should be taken into consideration, and the degree of impact mainly refers to the indicative requirements in Jilin Province or national relevant plans. In the process of transformation and upgrade of the Jilin Province, the stale development model of traditional industries has not been completely broken, while the emerging development model of modern industry has not been entirely established, leading to relatively backward economic and technological development. Therefore, BAU is the maximum boundary that can be reached when various factors promoting economic development are fully exploited and ecological and environmental benefits are less considered, which is basically consistent with the performance of historical development.

3.2.2. Energy-Saving Scenario (ESS)

Compared with BAU, which is mainly oriented towards industrial economic development, energy-saving scenario (ESS) focuses on energy conservation and emission reduction. Based on striving to achieve the goal of industrial transformation, Jilin Province further formulate the reasonable requirements for energy conservation and emission reduction in the policy planning and improve the energy consumption structure and energy utilization efficiency from the structural and technical aspects. Specifically, the key factors that influence energy utilization, such as energy structure and energy intensity, strike a balance between historical development and planning goals, and then change the future development growth rate of energy utilization to achieve the goals in the future development process. In this scenario, based on the direct impact of energy on $CO_2$ emissions, it basically reflects the relationship between industrial development and $CO_2$ emission status under the guidance of energy-saving policies.

3.2.3. Energy-saving and Low-Carbon Scenario (ELS)

Compared with the energy-based ESS, energy-saving and low-carbon scenario (ELS) not only further improves the low-carbon technology and energy efficiency to meet the energy intensity requirements of the 13th Five-Year Plan, but also adjusts the future development growth rate of factors such as industrial structure and urbanization, which basically reflects the industrial development and carbon emission status that can be achieved under the promotion of energy-saving and low-carbon policies. In this scenario, the emphasis is placed on the coordinated development of industrial society and the ecological environment, not through the improvement of the ecological environment at the cost of the industrialization process, but through the progress of low-carbon technology and structural adjustment reflecting the high-quality development of industrialization.

3.2.4. Low-Carbon Scenario (LCS)

Compared to ELS, low-carbon scenario (LCS) has a certain duality. On the one hand, In LCS, the concept of low-carbon development has penetrated into all aspects of industrial development, which

is characterized by further accelerating the pace of structural adjustment, promoting the upgrading of urbanization level, optimizing the energy and industrial structure, and controlling the increase in population. On the other hand, although all these adjustments can significantly promote the decline of industrial $CO_2$ emissions, it mainly considers the development needs of a low-carbon society and to some extent ignores the development needs of the industry. Specifically, it only measures pros and cons of development via various indicators and refers to the development process and level of developed countries or regions, but it fails to effectively combine the actual development situation and future path selection of Jilin Province's industry and lacks the long-term mechanism for the development of low-carbon industries.

### 3.3. Coefficient Setting

In order to comprehensively consider the uncertainty caused by future development policy changes and technological progress in Jilin Province's industry, the influencing factors are set to high mode, medium mode, and low mode, and appropriate parameters need to be set.

For six factors, including industrial energy consumption $CO_2$ emission efficiency, industrial energy intensity, industrialization rate, GDP, urbanization rate and the share of renewable energy, from the perspective of emission reduction, the lower the value of each factor is, the smaller the regional $CO_2$ emission becomes. Therefore, each various factor value was divided into low, medium and high modes. The basis of setting the parameter is the correspondence of the main factors to a data set, but not a data point. For example, urbanization rate factors correspond to the data sets of the urban population in the total population. Since the settings are rather complex, each factor's rates of change were divided into high, medium and low changing modes.

### 3.3.1. Industrial Energy Consumption $CO_2$ Emission Efficiency

Industrial energy consumption $CO_2$ emission efficiency not only represents the factor of industrial energy consumption structure but also reflects the progress of industrial clean-coal and low-carbon technology. In terms of energy consumption structure, Jilin Province is a typical northeast old industrial base, the current cost, technology, and geographical conditions restrict the upgrading of energy consumption structure to a certain extent. As for the industrial structure, there is no doubt that the high energy-consuming industries need a longer transition period to complete the upgrading, which also limits the optimization of energy structure. Therefore, the value of carbon emission efficiency from industrial energy consumption has shown an increasing trend in the historical stage, which was consistent with the overall situation in China. This study set the continuation of historical development state to a low mode, which will increase this value to 3.4 $tCO_2$/tce in 2020 and then decrease it to 2.4 $tCO_2$/tce in 2050. Combined the energy consumption situation in Jilin Province in recent years and the method provided by IPCC guidance catalog, the medium mode is set to reach the average level around 2020 of the European Union (EU) developed countries by 2050, in which this value is about 2.0 $tCO_2$/tce; the high mode is set to reach the average level around 2030 of the EU developed countries by 2050, in which this value is about 1.5 $tCO_2$/tce [22]. As shown in Table 3.

### 3.3.2. Industrial Energy Intensity

Industrial energy intensity is the characterization of technological progress in industrial processes. The higher the efficiency of industrial energy use, the smaller the intensity of industrial energy is. In recent years, as the old industrial base, Jilin Province is significantly behind the southeast coast in the economic and technological development aspect and does not obviously decrease industrial energy intensity. The statistics show that the average change rate of industrial energy intensity is −3.4% from 2011 to 2015. The "Opinions on the Recent Support to the Significant Policy Initiatives for the Revitalization of the Northeast" issued in 2014 proposed to rely on endogenous development to promote the northeast economy [23]. Taking into account the policy of the country and the transformation of Jilin Province, the improvements in technology and industrial structure will inevitably lead to changes

in industrial energy intensity. Therefore, this study refers to the industrial energy intensity change rate during the period of 12th Five-Year Plan, setting the average annual change rate of high mode in 2016–2050 to −4%, and the energy intensity in 2050 to 0.22 tce/$10^4$ yuan. The average annual change rate between medium mode and high mode is −5% and −6% in 2016–2050, and the energy intensity in 2050 is 0.17 tce/$10^4$ yuan and 0.12 tce/$10^4$ yuan respectively. As shown in Table 3.

**Table 3.** The settings of various factors in the prediction of $CO_2$ emissions from industrial energy consumption.

| Influencing Factor | Mode | 2020 | 2030 | 2040 | 2050 |
|---|---|---|---|---|---|
| Industrial $CO_2$ emission efficiency (t$CO_2$/tce) | Low-mode | 3.35 | 3.00 | 2.69 | 2.41 |
| | Medium-mode | 2.97 | 2.63 | 2.33 | 2.02 |
| | High-mode | 2.85 | 2.33 | 1.90 | 1.55 |
| Industrial energy intensity (tce/ $10^4$ Yuan) | Low-mode | 0.55 | 0.35 | 0.27 | 0.22 |
| | Medium-mode | 0.49 | 0.29 | 0.21 | 0.17 |
| | High-mode | 0.43 | 0.25 | 0.16 | 0.12 |
| Industrialization rate (%) | Low-mode | 40.82 | 42.22 | 37.23 | 30.26 |
| | Medium-mode | 41.41 | 44.52 | 41.29 | 34.60 |
| | High-mode | 42.25 | 49.27 | 46.60 | 38.79 |
| GDP (Billion Yuan) | Low-mode | 1293.73 | 1996.88 | 2647.31 | 3103.52 |
| | Medium-mode | 1609.34 | 3028.14 | 4890.19 | 5984.33 |
| | High-mode | 1740.24 | 3631.37 | 6106.98 | 8265.52 |
| Urbanization rate (%) | Low-mode | 56.82 | 60.07 | 62.77 | 64.55 |
| | Medium-mode | 57.99 | 62.45 | 66.63 | 69.67 |
| | High-mode | 59.53 | 65.76 | 71.21 | 75.00 |
| The share of renewable energy (%) | Low-mode | 12.12 | 17.68 | 22.08 | 24.39 |
| | Medium-mode | 13.92 | 21.12 | 30.62 | 34.84 |
| | High-mode | 16.09 | 26.49 | 38.62 | 44.68 |

### 3.3.3. Industrialization Rate

The industrialization rate characterizes the degree of regional industrial development. When the level of industrial development is saturated, the industrialization rate will gradually decline. In most provinces of China, the peak of the industrialization rate is concentrated in the range of 40–50%. During the 12th Five-Year Plan period, the industrialization rate decreased from 47% to 44%, which shows a similar trend with the average level of China, but the driving force behind the trend is different between the Jilin Province and the other developed areas of China. The national average situation belongs to the long-term external performance because of the optimization and upgarde of the industrial structure and the adjustment based on the healthy and stable development of the economy. However, Jilin Province is still in the middle stage of industrialization and Jilin Province as an old industrial base in northeast China is difficult to finish the industrial transformation. In the process of slow development of emerging industries, traditional industries cannot continue to effectively provide economic development impetus, which leads to the decline of industrialization level in recent years. For these reasons, in the future transformation and development of Jilin Province, the industrialization rate will be expected to change from decrease to increase if emerging industries successfully rise. Under the low, medium and high modes, the industrialization rate will continually decline until around 2020, and then it will increase in various degrees. Among these modes, the low mode slowly will increase to 43% and then decrease, and the proportion in 2050 will reach 30%; the medium mode and the high mode will increase to 45% and 50% respectively, and then fall to 35% and 39% in 2050. As shown in Table 3.

### 3.3.4. GDP

GDP represents the development of regional economies. During the period of 12th Five-Year Plan, the GDP growth rate of Jilin Province in 2011 and 2012 was 13.7% and 13.8%, respectively, and decreased to 8.1% in 2013 and to 6.5% in 2014 and 2015. The average growth rate of GDP from 2010 to 2020 has been set to 8% in the 2050 China Energy and $CO_2$ emission Report [24], which is basically the same as the GDP growth rate of the Jilin Province in 2013. Meanwhile, the economic development goal that keeps a medium-to-high growth rate has been put forward in the 13th-Five Plan of Jilin Province. On this basis, the average annual GDP growth rate of 8% is set to medium mode between 2016 and 2020, 7% and 10% are set to low mode and high mode respectively. After that, it will drop by 0.5 percentage points every five years. By 2050, the GDP corresponding to the three modes is 310.35 billion Yuan, 598.43 billion Yuan, and 826.55 billion Yuan respectively. As shown in Table 3.

### 3.3.5. Urbanization Rate

The urbanization rate represents the scale of urban development. According to estimates by the United Nations, the urbanization rate of developed countries in the world will reach 86% in 2050, and China's urbanization rate will reach 71.2% in 2050. 2050 China Energy and $CO_2$ Emissions Report proposes that the urbanization rate of China will attain 63% by 2020 [24]. In addition, the New Urbanization Plan of Jilin Province (2014–2020) has mentioned that the urbanization rate of Jilin Province will achieve about 60% in 2020 [25]. Combined with the average rate of change in the urbanization rate of Jilin Province in recent years, this study sets the target of 60% reached around 2030 as the low mode, and the corresponding time of medium mode and high mode are set at 2025 and 2020 respectively. As shown in Table 3.

### 3.3.6. The Share of Renewable Energy

The share of renewable energy represents the level of renewable energy use in total energy consumption, characterizing the utilization degree of regional renewable energy. Sino–US Joint Statement on Climate Change announces that the share of renewable energy in China will reach more than 20% by 2030 [26]. Consisting with the development policy of renewable energy in the future, we set the target of 20% reached around 2030 as the medium mode; the target of low mode is to reach 20% by 2035; the high mode will account for about 45% by 2050. As shown in Table 3.

According to the suitable changing patterns of each factor (i.e., high-mode, medium mode, or low mode), the four scenarios were composed. As is shown in Table 4.

**Table 4.** The options of scenario's parameters.

| Scenario | Industrial $CO_2$ Emission Efficiency | Industrial Energy Intensity | Industrialization Rate | GDP | Urbanization Rate | The Share of Renewable Energy |
|---|---|---|---|---|---|---|
| BAU | Low | Low | High | High | High | Low |
| ESS | Medium | Medium | Medium | High | High | Medium |
| ELS | High | High | Medium | High | Medium | Medium |
| LCS | High | High | Low | Medium | Low | High |

Note: high, medium and low represent high mode, medium mode and low mode respectively.

## 4. Results and Discussion

The results show that $CO_2$ emissions from industrial energy consumption in Jilin Province increase first and then decrease in different scenarios, which is basically consistent with most research results. The peak value ranges from 147.87 Mt to 229.73 Mt, and the peak time ranges from 2026 to 2043. According to the tendency under the BAU, the peak time will occur in 2043 with a peak value of 229.73 Mt. Compared with studies in other regions, the peak time of this scenario appears significantly

later [10–12]. Due to factors such as advanced low-carbon technology, LCS will peak earlier than any other scenarios, at 147.87 Mt in 2026. ESS and ELS will peak in 2035 and 2030 with the peak value of 190.89 Mt and 160.94 Mt respectively, as shown in Figure 5. $CO_2$ emissions from industrial energy consumption exhibit a trend of slow growth before reaching the peak due to technological progress and energy-saving and emission-reduction policy. After the peak, carbon emissions show a rapid decline due to the innovation of industrial technology and the strengthening of relevant policies.

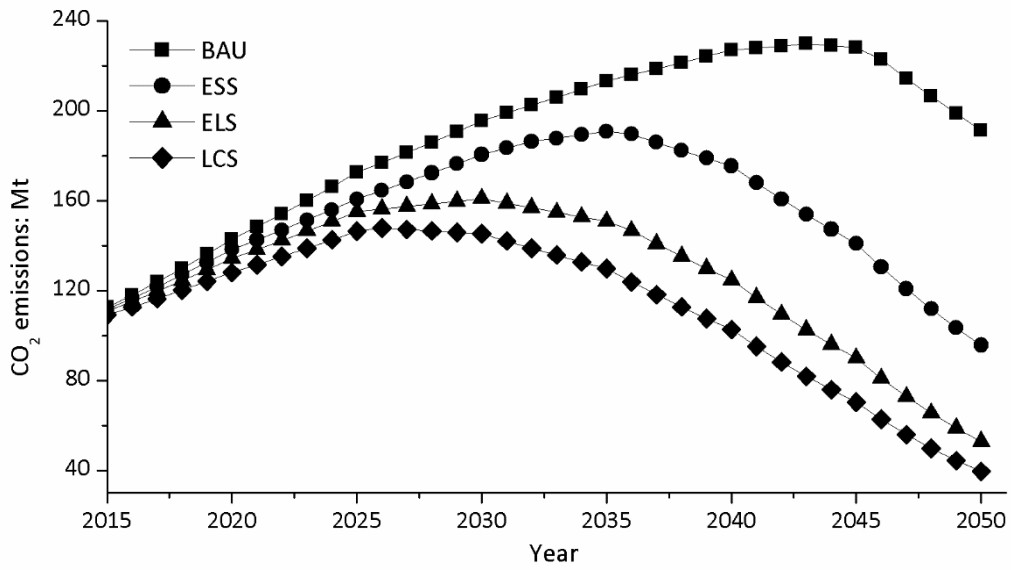

**Figure 5.** Predicting the results of Jilin Province's industrial $CO_2$ emissions in four scenarios.

The peak value of BAU is 229.73 Mt in 2043, which is 38.84 Mt, 68.79 Mt, and 81.86 Mt higher than ESS, ELS and LCS respectively, and the peak time of BAU also lags behind ESS, ELS, and LCS by 17 years, 13 years and 8 years severally. In addition, in the view of the total $CO_2$ emissions from 2015 to 2050, the cumulative carbon emissions from BAU, ESS, ELS and LCS are 6.81 billion tons, 5.55 billion tons, 4.56 billion tons and 4.05 billion tons respectively. Among them, the carbon emission of BAU is 1.7 times that of LCS, indicating that the later the peak time comes, the greater the cumulative $CO_2$ emission of industrial energy consumption will be. According to the model fitting result, if Jilin Province further optimizes the energy structure, industrial structure and energy efficiency in the industrial field, the peak time will come early and the peak value will be reduced significantly.

*4.1. Analysis of the Optimal Scenario*

Different scenarios characterize the diverse development trend of Jilin Province in areas of economy and technology. The judgment and selection of the optimal scenario is carried out based on reference to the corresponding indicators of the peak from $CO_2$ emissions of industrial energy consumption in developed countries and regions. Taking industrial energy intensity as an example, the energy intensity of BAU, ESS, ELS and LCS at the peak are 0.20 tec/$10^4$ Yuan, 0.18 tec/$10^4$ Yuan, 0.15 tec/$10^4$ Yuan and 0.14 tec/$10^4$ Yuan respectively. Statistics show that the average energy intensity level of EU reached the peak is 0.16 tec/$10^4$ Yuan [27], which is close to ELS. From the perspective of peak value, the industrial energy consumption $CO_2$ emissions peak value of BAU and ESS are approximately 1.6 times as much as the value in 2014, which is inconsistent with the requirement of the total national energy consumption control. The energy consumption and $CO_2$ emissions are both the least in LCS because of the improvement of industrial energy intensity, industrialization rate, the share of renewable energy and other aspects. However, its low-carbon technical requirements are so high that the technical level is difficult to achieve in a short period, otherwise, it may need to pay a higher social cost to meet the requirements of low-carbon development. In a stage of transition from ESS to LCS, the prediction results of ELS are basically in line with the general laws of industrial

development, and the peak of $CO_2$ emissions in 2030 coincide exactly with the year in which China promise to the world.

*4.2. Analysis of the Driving Factors of the Peak Value and Time*

In this paper, we used the method of control variables which only change the single-factor rate in turn under the premise of remaining other factors rate unchanged to quantitatively analyze the influencing direction and degree of driving factors to the $CO_2$ emissions peak. The results are shown in Table 5.

**Table 5.** $CO_2$ emissions peaks with different rates in each scenario.

| Factor | Rate | LCS | | ELS | | ESS | | BAU | |
|---|---|---|---|---|---|---|---|---|---|
| | | Peak Year | Peak Value | Peak Year | Peak Value | Peak Year | Peak Value | Peak Year | Peak Value |
| Industrial $CO_2$ Emission Efficiency | Low | 2035 | 178.80 | 2040 | 213.39 | 2040 | 215.08 | 2043 | 229.73 |
| | Medium | 2032 | 163.83 | 2035 | 189.46 | 2035 | 190.89 | 2035 | 194.51 |
| | High | 2026 | 147.87 | 2030 | 160.94 | 2030 | 162.10 | 2030 | 165.52 |
| Industrial Energy Intensity | Low | 2026 | 147.87 | 2030 | 156.25 | 2035 | 187.63 | 2043 | 229.73 |
| | Medium | 2026 | 145.99 | 2030 | 158.55 | 2035 | 190.89 | 2043 | 234.69 |
| | High | 2026 | 144.49 | 2030 | 160.94 | 2035 | 194.28 | 2043 | 239.85 |
| Industrialization Rate | Low | 2026 | 147.87 | 2026 | 152.39 | 2032 | 171.74 | 2035 | 186.08 |
| | Medium | 2030 | 155.04 | 2030 | 160.94 | 2035 | 190.89 | 2040 | 212.24 |
| | High | 2030 | 159.78 | 2030 | 165.86 | 2035 | 196.73 | 2043 | 229.73 |
| GDP | Low | 2026 | 143.19 | 2030 | 151.90 | 2035 | 177.62 | 2043 | 209.04 |
| | Medium | 2026 | 147.87 | 2030 | 158.36 | 2035 | 187.00 | 2043 | 223.28 |
| | High | 2026 | 150.19 | 2030 | 161.26 | 2035 | 190.89 | 2043 | 229.73 |
| Urbanization Rate | Low | 2026 | 147.87 | 2030 | 158.52 | 2035 | 183.14 | 2043 | 217.93 |
| | Medium | 2026 | 149.71 | 2030 | 160.94 | 2035 | 186.55 | 2043 | 223.54 |
| | High | 2026 | 152.10 | 2030 | 164.21 | 2035 | 190.89 | 2043 | 229.73 |
| The Share of Renewable Energy | Low | 2026 | 149.14 | 2030 | 161.81 | 2035 | 192.01 | 2043 | 229.73 |
| | Medium | 2026 | 148.40 | 2030 | 160.94 | 2035 | 190.89 | 2043 | 228.26 |
| | High | 2026 | 147.87 | 2030 | 160.27 | 2035 | 190.01 | 2043 | 227.04 |

Note: high, medium and low represent high mode, medium mode and low mode respectively.

4.2.1. Industrial $CO_2$ Emissions Efficiency

In the four scenarios, Jilin Province's peak value of industrial energy consumption with a low rate of $CO_2$ emissions efficiency is 30.93 Mt to 64.21 Mt, which is higher than the peak value with a high rate. That is to say, the industrial energy consumption $CO_2$ emission efficiency is an effective $CO_2$ reduction factor affecting the peak of $CO_2$ emissions. The higher the $CO_2$ emissions efficiency change rate of industrial energy consumption is, the earlier the peak appears. Taking ELS as an example, the peak time with a high rate of industrial energy consumption $CO_2$ emissions efficiency is 2030, and the peak time with a low rate is 2040, 10 years in advance. The result is in line with the ridge regression equation fitted with a high elasticity coefficient of 1.38, which shows that the industrial energy consumption $CO_2$ emissions efficiency has a significant effect on the peak time and value. The result shows that the industrial energy consumption $CO_2$ emissions efficiency is a significant factor to push carbon reduction. The essential driving factors to change the $CO_2$ emissions efficiency lies in the improvement of energy-efficient and exploitation of renewable energy.

4.2.2. Industrial Energy Intensity

During the period of 2001 to 2015, the average change rate of industrial energy intensity is −8% in Jilin Province, while the average change rate of industrial energy intensity is −9% during the period of 12th Five-Year Plan, which presents the trend of continuous decline. The higher the rate of decline in industrial energy intensity is, the smaller the peak value is. By changing the rate of industrial energy

intensity, Jilin Province's peak value of the industrial energy consumption $CO_2$ emissions with a low rate is 3.38 Mt to 10.12 Mt, which is higher than the peak value with a high rate in the four scenarios. The comparative analysis shows that the industrial energy intensity at the present stage is a factor of $CO_2$ emission reduction, but the effect degree on peak value is significantly less than that of the $CO_2$ emission coefficient of industrial energy consumption and the minimum value is only one-tenth of it. Compare to the $CO_2$ emissions coefficient, the industrial energy intensity has no effect on the peak time and has a relatively small impact on the peak value.

### 4.2.3. Industrialization Rate

In the four scenarios, the peak with a high rate is 11.91 Mt to 43.65 Mt, which is higher than the peak value at a low rate. That is to say, the industrialization rate is an obvious carbon increase factor. The industrialization rate has a greater impact on peak time. However, in contrast to the effect of $CO_2$ emission efficiency of industrial energy consumption, the higher the industrialization rate is, the later peak time is. Taking ELS as an example, the peak time with a high rate of industrialization rate is 2030, but the peak time with a low rate is 2026, a gap of 4 years. In addition, the coefficient of the standard ridge regression equation demonstrates that the industrialization rate has a higher degree of influence on the industrial energy consumption $CO_2$ emissions in Jilin Province, which also reflects the industry is the main sector of energy consumption. With other factors remaining unchanged, the change rate based on the original industrialization rate is basically consistent with the elastic coefficient (0.83).

### 4.2.4. GDP

The results show that the peak value with a high rate of GDP is 7.00 Mt to 20.69 Mt, which is higher than the peak value at a low rate by changing the rate of GDP. GDP is a typical carbon increase factor. Besides, GDP has a significant impact on the peak. Taking ELS as an example, the peak value at the high rate of GDP is 13.28 Mt, which is higher than the peak value with a low rate, indicating that GDP is the carbon increase factor for industrial energy consumption in Jilin Province. Compared to the industrial energy intensity, GDP does not affect peak time in four scenarios no matter how much the rate changes.

### 4.2.5. Urbanization Rate

The peak value with a high rate of Urbanization rate is 4.22 Mt to 11.80 Mt, which is higher than the peak value with a low rate, demonstrating that the development stage of urbanization is accompanied by people from rural production and lifestyle to urbanization production and lifestyle. Energy consumption of infrastructure construction and demand for household electrical appliances directly or indirectly affect the $CO_2$ emissions change of industrial energy consumption. By contrast, the impact of urbanization on $CO_2$ emissions of industrial energy consumption is not as significant as the industrialization rate. In addition, the urbanization rate changed to three different rates and the peak time in four scenarios are all the same year, indicating that the urbanization rate has no effect on the peak time of industrial energy consumption carbon emissions.

### 4.2.6. The share of Renewable Energy

The results show that the share of renewable energy is the carbon reduction factor, but the effect is not significant. The peak value with a low rate is 1.27 Mt to 2.69 Mt, which is higher than the peak value with a high rate. The peak time of each scenario is the same year even if the ratio of renewable energy consumption is changed. Taking ELS as an example, the peak time of industrial energy consumption $CO_2$ emissions in Jilin Province is 2030, no matter how the renewable energy ratio changed, the renewable energy ratio does not affect the peak time. As Jilin Province is still in the stage of rapid economic development, fossil energy consumption accounts for a large proportion of total energy consumption, and there is less consumption of renewable energy due to the constraints of cost and technology conditions.

*4.3. Analysis of the Reduction Pathway of $CO_2$ Emissions Peak Value and Time*

4.3.1. Analysis of the Reduction Pathway of Peak Value

The six factors, including industrial energy consumption $CO_2$ emissions coefficient, GDP, industrialization rate, urbanization rate, industrial energy intensity and share of renewable energy, have diverse degrees of influence on industrial carbon emissions peak value. Taking ELS as an example, the influencing degree of six factors in ELS are: industrial energy consumption $CO_2$ emission efficiency (1.33) > industrialization rate (1.09) > GDP (1.06) > urbanization rate (1.04) > industrial energy intensity (1.03) > the share of renewable energy (1.01), as shown in Figure 6, which is also consistent with the result in Table 5. The $CO_2$ emissions coefficient of industrial energy consumption has the most significant influence on the peak value, displaying of the effectiveness of the industrial energy structure adjustment and clean-coal and low-carbon technology development on the peak value. Followed by the industrialization rate, it is an effective measure to control carbon emissions by steadily promoting the development of industrialization and industrial upgrade. In addition, the contribution of other factors to the peak value has little difference. This result is quite different from the results of previous studies such as Zhang et al. [7] and Liu et al. [8] which shows that economic growth and energy structure have a more significant effect, indicating that the transition region has special characteristics in the transition stage.

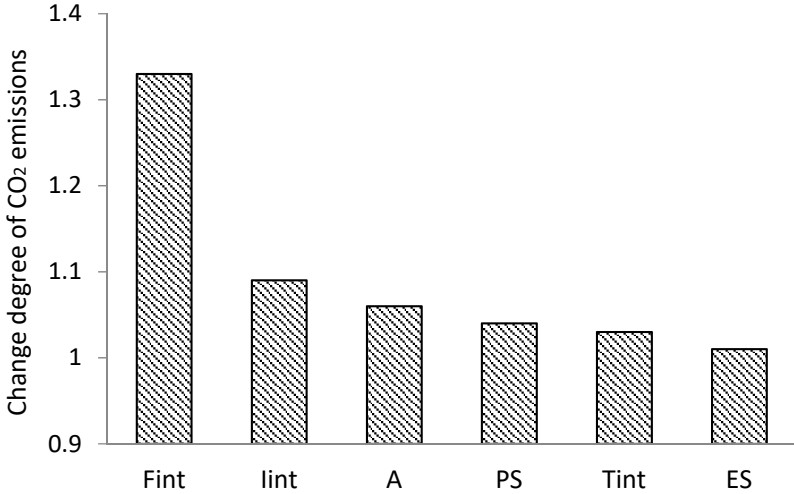

**Figure 6.** The influencing degree of factors on $CO_2$ emissions in energy-saving and low-carbon scenario (ELS).

In ELS, the peak value of the influencing factors from low to high rate shows that the value decreases from 213.39 Mt to 160.93 Mt when the industrial energy consumption $CO_2$ emission efficiency fluctuates from low mode to high mode. The change in peak value is −52.46 Mt, which represents the drive of the negative. When the industrial energy intensity changes from low mode to high mode, the peak value decreases from 160.93 Mt to 156.25 Mt, and the change is −4.68 Mt. Similarly, when the share of renewable energy changes from low mode to high mode, the peak value decreases from 161.81 Mt to 160.27 Mt and the change value is −1.54 Mt. The influence degree of this factor is 2.93% of the industrial energy consumption $CO_2$ emissions efficiency. In addition to the above two factors, the other three factors are all positive driving factors. The peak change value of industrialization rate, GDP and urbanization rate from low mode to high mode are 13.46 Mt, 9.36 Mt and 5.69 Mt respectively and the influence degree of these factor are 25.68%, 17.85% and 10.85% of the industrial energy consumption $CO_2$ emissions efficiency. As shown in Figure 7.

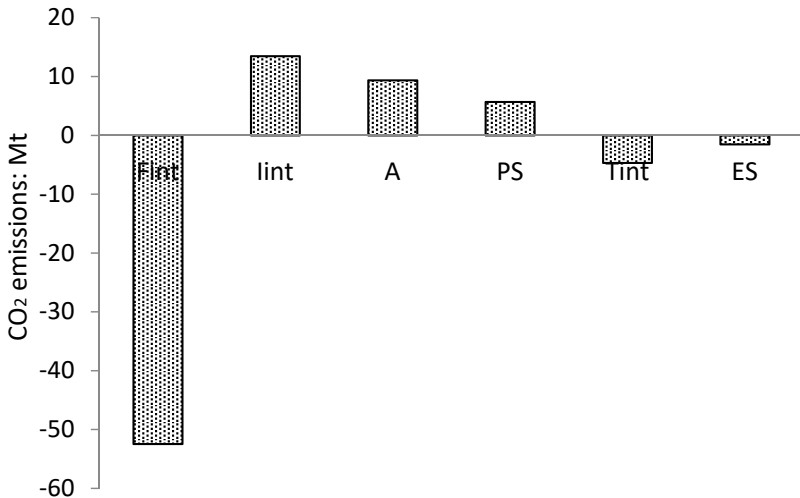

**Figure 7.** The peak value difference with variation rate changing in ELS.

#### 4.3.2. Analysis of the reduction pathway of Peak Time

Analysis of the influencing degree of industrial energy consumption $CO_2$ emissions from the perspective of the peak proves that the peak will occur when the carbon increase effect and the carbon reduction effect of the influencing factors in each scenario. When the rate of factors changes, the new balance will replace the previous balance, and the peak time also changes accordingly. Only two main factors do effect on the peak time including industrial energy consumption $CO_2$ emission efficiency and industrialization rate influencing the peak time, while the others do not affect the peak time. The higher the rate of industrial energy consumption $CO_2$ emission efficiency, the earlier the peak time appears. On the contrary, the higher the rate of industrialization rate, the later the peak time appears, which is similar to the study by Ge et al. [13]. Taking ELS as an example, the peak time well ahead of 10 years when the industrial energy consumption $CO_2$ emission efficiency changes from low mode to high mode, while the peak time will delay four years when the industrialization rate changes from low mode to high mode. Therefore, it is an effective means to achieve the goal of earlier realization of industrial energy consumption $CO_2$ emissions peak by reducing industrial energy consumption $CO_2$ emission efficiency and the change rate of industrialization rate. In contrast, other factors only affected the peak value.

#### 4.4. Analysis of the Reduction Mechanism of $CO_2$ Emissions Peak Value and Time

Jilin Province responds to national targets and plans to reach its $CO_2$ emissions peak by 2030. However, since the level of socio-economic development in Jilin Province lags the average level in China, the $CO_2$ emissions peak time in Jilin Province may be delayed due to the lack of financial and technological support. In order to achieve the peak of overall $CO_2$ emissions in advance, it is particularly important to reach the industrial carbon emissions peak as soon as possible. Overall, ELS is the optimal scenario choice of development to Jilin Province compared to the other three scenarios, which is feasible for the future development of Jilin Province. Because it can achieve a better status of $CO_2$ emissions peak under the premise of ensuring stable economic and social development. Then, how does transform the BAU with a peak time of 2043 and a corresponding peak value of 229.73 Mt into the ELS with a peak time of 2030 and a corresponding peak value of 160.94 Mt? Figure 8 shows the relevant emission reduction mechanism from the perspectives of technological effect, structural effect and scale effect.

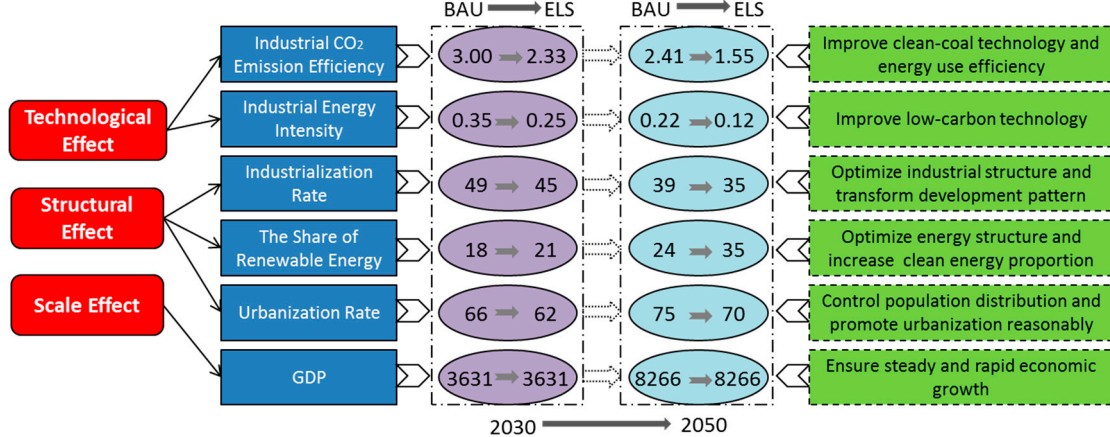

**Figure 8.** Carbon reduction mechanism and roadmap of baseline scenario (BAU) to ELS.

Specifically, by 2030 and 2050, industrial energy consumption $CO_2$ emissions efficiency will decrease from BAU's 3.00 $tCO_2$/tce and 2.41 $tCO_2$/tce by 22% and 36%, respectively, to ELS's 2.33 $tCO_2$/tce and 1.55 $tCO_2$/tce. For industrial energy intensity, it will decrease from BAU's 0.35 tce/ $10^4$ Yuan and 0.22 tce/$10^4$ Yuan by 29% and 45%, respectively, to 0.25 tce/ $10^4$ Yuan and 0.12 tce/ $10^4$ Yuan of ELS. For the industrialization rate, it will be reduced from BAU's 49% and 39% by 10% and 11%, respectively, to 45% and 35% of ELS. For the share of renewable energy, it will increase from BAU's 18% and 24% by 19% and 43%, respectively, to 21% and 35% of ELS. For the urbanization rate, it will decrease from 66% and 75% of BAU by 5% and 7%, respectively, to 62% and 70% of ELS. Considering the importance of economic development, GDP growth will remain basically the same, reaching 3631 billion Yuan and 8266 billion Yuan by 2030 and 2050 respectively. As shown in Figure 8.

### 4.5. Policy Implications

Based on the above research results, the following policy implications are proposed:

(1) In order to control the $CO_2$ emissions peak value of industrial energy consumption, it's essential to refer to the development mode of economy, technology, and structure in ELS. The better controllable path are: Firstly, vigorously develop clean-coal and low-carbon technologies to reduce the value of $CO_2$ emissions efficiency of industrial energy consumption; Secondly, adjust the industrial structure on the premise of stabling economic development; Thirdly, other factors such as economic development, urbanization development, industry energy intensity and the share of renewable energy, due to the small degree of impact, can take certain measures, but the effect of the first two controllable paths are relatively obvious.

(2) The main paths to realize the peak time of industrial energy consumption $CO_2$ emissions earlier are: firstly, reduce the $CO_2$ emission efficiency of industrial energy consumption, and vigorously develop clean-coal technology and extensively apply low-carbon technologies; secondly, carry out adjustment of industrial structure and control the development of industrialization rate on the premise of stabling economic growth.

## 5. Conclusions

From the perspective of peak value, we proposed the extended STIRPAT model to analyze the reduction path of industrial energy consumption $CO_2$ emissions in Jilin Province. Based on the simulation result of BAU, ESS, ELS and LCS, the peak time of industrial energy consumption $CO_2$ emissions in Jilin Province are 2043, 2035, 2030 and 2026 respectively, and the corresponding peak value are 229.73 Mt, 190.89 Mt, 160.94 Mt, and 147.87 Mt. Among them, ELS reached the peak in 2030, which is basically consistent with the voluntary contribution target proposed by China after the Paris Agreement. Overall, ELS is the optimal scenario to reduce industrial energy consumption

$CO_2$ emissions. The results show that the industrial energy consumption structure, industrial energy intensity and the share of renewable energy are the reducing carbon factors while GDP, industrialization rate and urbanization rate are the increasing carbon factors, which is basically the same with the research conclusion drawn by Liu et al. [8]. The six influencing factors all have an impact on the peak value of industrial energy consumption $CO_2$ emissions, but the industrial energy consumption structure and the industrialization rate have a higher influencing degree. In contrast, peak time is only affected by two factors including industrial energy consumption carbon emission efficiency and industrialization rate, and other factors do not impact peak time. The influencing degree of six factors in ELS are: Industrial energy consumption $CO_2$ emissions efficiency (1.33) > industrialization rate (1.09) > GDP (1.06) > urbanization rate (1.04) > industrial energy intensity (1.03) > the share of renewable energy (1.01). Therefore, the main paths to realize the lower peak value of industrial carbon emissions earlier are to adjust the structure of energy consumption and industry and properly control the growth of industrialization on the premise of stabling economic development. If northeast China expects to peak by 2030, the relevant emission reduction mechanism from the perspectives of technological effect, structural effect and scale effect need to be done, such as the industrial energy consumption $CO_2$ emissions efficiency need decrease from BAU's 3.00 tCO2/tce and 2.41 tCO2/tce. The economic and social development needs to follow the ELS scenario.

Through scenario setting and path analysis, our study can provide references for achieving $CO_2$ emissions peak from industrial energy consumption in typical transformation regions. However, due to the limitations of the model and the complexity of socio-economic development, the STIRPAT model may not take all influencing factors into account, coupled with the uncertainty of future development, there may be a certain degree of deviation between our prediction results and the actual situation in the future. Meanwhile, we take Jilin province as a typical case, which may lack extensive representativeness, therefore, in the following research, we will further improve the model and take other transformation regions as examples for relevant exploration.

**Author Contributions:** Z.D. designed the research work, carried out the data collection and wrote the paper. X.W. performed the data checking, results analysis. J.S., S.W. and X.D. did the language modification. H.D. and J.S. provided academic advice throughout the process. All authors have read and approved the published version of the manuscript.

**Funding:** This research was funded by the National Natural Science Foundation of China (No. 71773034, No. 41801199, No. 71704157) and 111 Project of Jilin University (No. B16020).

**Acknowledgments:** We would like to thank the editor and reviewers for their constructive comments and suggestions on the manuscript.

**Conflicts of Interest:** The authors declare no conflict of interest.

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
