# Peer review of "Peaking Industrial Energy-Related CO2 Emissions in Typical Transformation Region: Paths and Mechanism"

_sustainability, doi:10.3390/su12030791_

Round 1

Reviewer 1 Report

My personal assessment on the work is positive, owing to the quality and clarity of the study, as well as the focus of analysis. So, the study is interesting and results may have political implications.

But I believe it is necessary to undertake a set of modifications, in order to improve the current manuscript, to fulfil the standards of quality in such a prestigious journal. 

-The structure is not appropriate. The paper is divided into five parts: 1) an introduction; 2) the methodology; 3) Case Study 4) Results and Discussion; 5) conclusions. However, the introduction is very long compared, for example, to the study method and Data and the conclusions.

-It is necessary to expand the working hypotheses put forward and the specific research problems. This suggestion is made to assist less technically interested readers.

-More information is required on the definition of the indicators selected and need further explanation in relation to the period covered by these indicators.

- The authors need to include more theoretical underpinnings to their work and draw out the policy implications of the results much more clearly.

-Because of the lack of a proper current literature review the results are not discussed in the context of previous findings.

Due to the complexity of the model the theoretical and empirical underpinnings of the model specification are at best can be improved.

-Some practitioner readers of Sustainability wish to concentrate on the findings of the paper and the recommendations for policy. The policy implications of the analysis need to be made much more clear.

Reviewer 2 Report

The paper deals with important environmental problems in industrial sector of China and addresses climate change mitigation issues. But the paper needs revision and clarifications. The introduction should be rewritten. The figures and situation in China should be removed or shortened. The structure of paper should be given in the end of introduction. Another problem is not clear definitions. "The share of renewables". What kind of share? It should be clearly defined: share of renewables in final energy or share of renewables in primary energy or in electricity consumption etc. Why do this paper analyse CO2 emissions but not total GHG emission from energy consumption in industry? Why do  GHG emissions from industrial processes are not taken into account?

The paper should provide clarifications. In discussion of results the results of other studies in China  should be compared. There are many such type of studies. The strengths and limitations of applied approach should be addressed and future research guidelines should be provided. The policy implications of conducted study are also need to be improved. 

Round 2

Reviewer 1 Report

Overall Recommendation Report 2: Accept in present form

Reviewer 2 Report

The authors have improved paper and have taken all my comments into account. Just I would recommend  authors to avoid so many questions putted now in introduction. It is ok but maybe one question is enough.

Response: Following the reviewer's suggestion, we have summarized the original two questions into one in introduction, that is “Overall, what should be done to achieve the CO2 emissions peak of industrial energy consumption by reducing the CO2 emissions in typical transformation regions such as northeast China?”